# Possible Role of Amyloidogenic Evolvability in Dementia with Lewy Bodies: Insights from Transgenic Mice Expressing P123H β-Synuclein

**DOI:** 10.3390/ijms21082849

**Published:** 2020-04-19

**Authors:** Masayo Fujita, Gilbert Ho, Yoshiki Takamatsu, Ryoko Wada, Kazutaka Ikeda, Makoto Hashimoto

**Affiliations:** 1Addictive Substance Project, Tokyo Metropolitan Institute of Medical Sciences, 2-1-6 Kamikitazawa, Setagaya-ku, Tokyo 156-8506, Japan; ikeda-kz@igakuken.or.jp; 2PCND Neuroscience Research Institute, Poway, CA 92063 USA; giho@pcndneurology.com; 3Laboratory of Parkinson’s Disease, Tokyo Metropolitan Institute of Medical Sciences, 2-1-6 Kamikitazawa, Setagaya-ku, Tokyo 156-8506, Japan; takamatsu-ys@igakuken.or.jp (Y.T.); wada-rk@igakuken.or.jp (R.W.)

**Keywords:** Dementia with Lewy bodies (DLB), α-synuclein (αS), β-synuclein (βS), P123H βS, transgenic (Tg) mice, amyloidogenic evolvability, therapy

## Abstract

Dementia with Lewy bodies (DLB) is the second most prevalent neurodegenerative dementia after Alzheimer’s disease, and is pathologically characterized by formation of intracellular inclusions called Lewy bodies, the major constituent of which is aggregated α-synuclein (αS). Currently, neither a mechanistic etiology nor an effective disease-modifying therapy for DLB has been established. Although two missense mutations of β-synuclein (βS), V70M and P123H, were identified in sporadic and familial DLB, respectively, the precise mechanisms through which βS mutations promote DLB pathogenesis remain elusive. To further clarify such mechanisms, we investigated transgenic (Tg) mice expressing P123H βS, which develop progressive neurodegeneration in the form of axonal swelling and non-motor behaviors, such as memory dysfunction and depression, which are more prominent than motor deficits. Furthermore, cross-breeding of P123H βS Tg mice with αS Tg mice worsened the neurodegenerative phenotype presumably through the pathological cross-seeding of P123H βS with αS. Collectively, we predict that βS misfolding due to gene mutations might be pathogenic. In this paper, we will discuss the possible involvement of amyloidogenic evolvability in the pathogenesis of DLB based on our previous papers regarding the P123H βS Tg mice. Given that stimulation of αS evolvability by P123H βS may underlie neuropathology in our mouse model, more radical disease-modifying therapy might be derived from the evolvability mechanism. Additionally, provided that altered βS were involved in the pathogenesis of sporadic DLB, the P123H βS Tg mice could be used for investigating the mechanism and therapy of DLB.

## 1. Introduction

Dementia with Lewy bodies (DLB) is the second most prevalent neurodegenerative dementia after Alzheimer’s disease (AD) [1]. A histological hallmark of DLB is the formation of Lewy bodies, which contain aggregates of α-synuclein (αS), a presynaptic protein exhibiting amyloidogenic properties [2]. Accordingly, DLB is clinically classified as a α-synucleinopathy, which also includes Parkinson’s disease (PD) and multiple system atrophy (MSA) [2]. DLB is associated with several core symptoms, including fluctuating cognition [3], alertness or attention, visuospatial deficits and prominent visual hallucinations [4], REM sleep behavior disorder [5], and cardinal features of parkinsonism [6]. At present, no effective disease-modifying therapy is available for DLB.

Missense mutations of disease-relevant amyloidogenic proteins (APs) can be rarely associated with clinical neurodegeneration in humans. For instance, in DLB, one missense mutation of α-synuclein (αS), E46K, was identified in familial DLB [7], while two missense mutations of β-synuclein (βS), V70M and P123H, were found in sporadic and familial DLB, respectively [8]. Curiously, autopsy tissue from a P123H βS patient exhibited typical αS pathology without evidence of βS aggregation, suggesting that P123H βS might be involved in the stimulation of αS aggregation by unknown mechanisms [8]. Furthermore, neurodegenerative features such as lysosomal inclusions were identified in neurons expressing mutant βS [9,10]. Thus, a better understanding of the pathogenic role of βS mutations in vivo might lead to elucidation of the mechanism of DLB.

Recently, we proposed that neurodegeneration in aging may be attributed to amyloidogeioc evolvability [11,12,13]. In this context, the main objective of this paper is to discuss the possible involvement of amyloidogenic evolvability in the pathogenesis of DLB using transgenic (Tg) mice expressing P123H βS which developed progressive neurodegeneration [14]. We suggest that the stimulation of αS evolvability by P123H βS may underlie neuropathology of the P123H βS Tg mice. Furthermore, given the findings, a P123H βS Tg mouse model could be used to develop radical, but perhaps more effective, therapies for DLB based on αS evolvability. If altered βS is involved in the pathogenesis of sporadic DLB, then P123H βS Tg mice might represent a generalized model for investigating DLB mechanisms and therapy.

## 2. Comparison of the P123H βS Tg Mice with DLB

At present, numerous PD and other α-synucleinopathy models exist where αS cDNA is overexpressed under various promoters [15]. Yet, as far as is known, only a few if any animal models exist specific to DLB.

### 2.1. P123H βS Tg Mice

In our study, Tg mice were created using Thy-1 promoter to drive familial DLB-derived P123HβS cDNA (Figure 1a) [14]. The P123HβS Tg mice developed progressive neurodegeneration, and were histologically characterized by formation of axonal swelling and small spheroids designated as “globules” that frequently contain small vesicles and membranous structures (Figure 1b,c). Indeed, formation of axonal spheroids was described in α-synucleinopathy brain tissue, including DLB and neurodegeneration with brain iron accumulation, type 1 (NBIA type 1) [16,17]. This suggests that βS might be involved in axonal pathology in the neuronal soma.

Similar to human DLB patients, P123H βS Tg mice showed behavioral abnormalities, with the non-motor symptoms being more prominent than motor deficits [14] (Figure 2). Whereas motor deficits become apparent after approximately 12 months as assessed by rota-rod test (Figure 2a), memory dysfunction was apparent at approximately 6 months of age on the Morris water maze test (Figure 2b). Furthermore, spontaneous activity was reduced at around 6 months of age as evaluated by home-cage tests (Figure 2c). Similarly, the mice exhibited hyperlocomotor activity in a novel environment at age 6 to 10 months as assessed from locomotor activity testing (Figure 2d). Finally, the mice exhibited depression-like behaviors manifested by reduced mobility time in the tail suspension test (Figure 2e) and impaired nest building [18]. In parallel to the above findings, REM sleep behavior disorder in DLB patients may be an early symptom that precedes memory loss and other features [5]. Furthermore, other frequent symptoms include visual hallucinations, marked fluctuations in attention or alertness, slowness of movement, trouble walking, and rigidity [1,3,4]. The autonomic nervous system is usually affected, resulting in altered blood pressure, heart function, and gastrointestinal function, with constipation as a common symptom [19]. Meanwhile, mood changes, such as depression and apathy, are also common [20].

To investigate the combined effect of P123H βS and αS, P123H βS Tg mice were subjected to cross-breeding with αS Tg mice [14,21]. The resulting bigenic (P123H βS/αS) mice exhibited more significant neurodegenerative phenotypic features when compared to P123H βS single Tg mice (Figure 3). In bigenic mice, both P123H βS and αS accumulated in degenerating neurons in the hippocampus and cerebral cortex which co-localized with each other (Figure 3a,b), suggesting that the cross-seeding of these APs may be central to the degenerative phenotype of the bigenic mice. Furthermore, severe motor impairments were already observed at 4 months old, as assessed by hind and front limb clasping (Figure 3c) and rota-rod test (Figure 3d). Consistent with these results, striatal dopamine concentrations were significantly reduced in the bigenic mice (Figure 3e), accompanied by a decrease in expression levels of dopaminergic markers such as tyrosine hydroxylase, L-dopa decarboxylase and dopamine transporter [14]. Interestingly, because of the lack of Lewy-body-like intraneuronal inclusions in both P123H βS Tg mice and bigenic mice, we speculate that both motor- and non-motor symptoms in Lewy body disorders might actually occur regardless of Lewy bodies. Alternatively, Lewy body formation may require a protracted timeframe to occur, and are absent in our mouse model due to their short lifespan. Nonetheless, although challenging to generate, we assert that the bigenic mice model is a more realistic paradigm for Lewy body diseases compared to the singly-transgenic P123H βS mouse.

Collectively, many behavioral abnormalities in both single P123H Tg mice and bigenic mice manifested earlier compared to formation of inclusion bodies. These results may suggest that small aggregates, including oligomers and protofibrils of APs, rather than mature fibrils of APs, might be attributed to behavioral alterations. Alternatively, protein aggregation of APs might be not related to alteration of behaviors in neurodegeneration.

### 2.2. Other DLB Model Mice

DLB has overlapping features of both AD and PD [22] and might be different from other similar diseases, including Lewy body dementia, Lewy body variant of AD, diffuse Lewy body disease and cortical Lewy body disease [23,24]. Notably, Masliah et al. performed a cross experiment of a Tg mouse expressing αS with a Tg mouse expressing amyloid precursor protein (APP) [25]. The double Tg mice had severe deficits in learning and memory, developed motor deficits before αS singly Tg mice, and showed prominent age-dependent degeneration of cholinergic neurons and presynaptic terminals. Furthermore, the bigenic mice had more αS-immunoreactive inclusions than αS singly Tg mice [25]. These results suggest that stimulation of αS aggregation by Aβ may be one pathogenic mechanism for Lewy body disease. Although the pathogenic mechanism of the αS/APP bigenic mice is clear in this case, the complexities of Lewy body disease mechanisms in humans, and the difficult and time-consuming experimental procedures to create bigenic model mice compared to single Tg mice, indicate some disadvantages for this paradigm.

More recently, Tg mice expressing familial DLB-linked E46K αS were also reported [26]. Although E46K αS mice developed Lewy-like and tau pathology, and were associated with motor impairments in an age-dependent manner, they lacked for the most part the presence of non-motor symptoms essential to the DLB phenotype [26]. Collectively, the P123H βS Tg mouse model is likely a more accurate representation of DLB, especially for investigating DLB non-motor symptoms.

## 3. Mechanism of DLB

It is expected that a better understanding of the pathogenic mechanism of neurodegeneration may lead to development of evidence-based therapy for neurodegenerative diseases, including DLB. Although clinical trials have been extensively performed based on several hypotheses for the mechanism of neurodegeneration, the results are so far unsatisfactory.

### 3.1. Conventional View of the Mechanism of Neurodegeneration

Currently, most prevailing views suggest that neurodegeneration may be attributed to the accumulation of the neurotoxic aggregates of amyloidgenic proteins. In this context, “amyloid cascade hypothesis” (ACH), a dominant hypothesis in AD, postulates that aggregation of Aβ triggers a toxic cascade of events, including aberrant phosphorylation of tau, neuritic plaque and neurofibrillary tangle formation, synapse loss, and neuroinflammation, eventually leading to neuronal death [27]. According to ACH, it is expected that reduction of Aβ could be beneficial for the therapy of AD. Based on such a notion, clinical trials, such as β- and γ-secretase inhibitors and Aβ immunotherapy, have been extensively performed [28]. None of them, however, have proved successful, raising a concern regarding the validity of the ACH.

In the meantime, much attention had been paid to the role of other factors, such as inflammation and oxidative stress, in the pathogenesis of aging-associated neurodegenerative diseases. In support of the former, several epidemiological studies have shown that nonsteroidal anti-inflammatory drugs may reduce the risk of development of AD and PD. However, clinical studies of ibuprofen and other structurally related compounds for AD have been unsuccessful [29,30]. In the P123H βS Tg mouse, we observed that buprofen ameliorates protein aggregation and astrocytic gliosis, but not cognitive dysfunction, suggesting that there is a gap between protein aggregation and cognitive function [31]. Furthermore, oxidative stress was another interest in terms of the mechanism and therapy of neurodegenerative diseases. Indeed, a number of reports of preclinical studies, including ours [32], previously showed the beneficial effects of the inhibition of oxidative stress on dysregulation of mitochondria in neurodegeneration. However, clinical studies of vitamin E for AD failed to demonstrate the beneficial effects of antioxidants [33].

### 3.2. Recent Hypothesis of the Mechanism of Neurodegeneration

One may therefore consider that a better understanding of the physiological functions of APs is necessary for successful therapy development in neurodegenerative diseases. In this context, it is noteworthy that Aβ was rapidly seeded by herpes simplex virus (HSV) 1 to protect against brain infection in cellular and Tg mice models of AD, suggesting that fibrillation of Aβ might play a protective role in innate immunity of the nervous system [34]. Consistent with this notion, many studies have recently shown an association between virus infection and neurodegenerative disorders, not only HSV1 in AD, but also an influenza virus in PD [35]. Thus, further investigations are warranted to investigate a possible association of DLB with virus infection.

Since elucidation of both pathological and physiological functions of APs might be critical for developing effective therapy for neurodegenerative diseases, we recently proposed that evolvability is a physiological function of APs. The concept of evolvability is based on the evolvability of yeast prion that is critical for yeast surviving in the stressful environment [11]. Furthermore, neurodegenerative diseases, including DLB, were interpreted as a manifestation of evolvability through antagonistic mechanisms during aging [12].

## 4. DLB and Amyloidogenic Evolvability

As with related conditions, we have posed the question as to why such neurodegenerative conditions as DLB might not only appear but persist across evolutionary pressures. In this regard, our recent view regarding amyloidogenic evolvability might provide a clue to address this question. Regarding the physiological function of APs relevant to neurodegeneration including αS, we recently proposed that amyloidogenic evolvability might be important in the brain exposed to multiple stressors, including hyperthermia, oxidative stress, and neurotoxicity [11]. Precisely, the diverse β-sheet structures of various protofibrillar APs might confer resistance against diverse stressors in parental brains, which may be transmitted to offspring through germ cells [36]. By virtue of the stress information derived from parental brains, an offspring’s brain can better cope with forthcoming stresses that alternatively would lead to the onset of neurodevelopmental disorders. On the other hand, neurodegeneration may manifest in the parental brain through the antagonistic pleiotropy mechanism in aging (Figure 4a) [12].

### 4.1. Familial DLB

Considering the missense mutations in αS, following discovery of an A53T substitution in PD [37], two other pathogenic substitutions (A30P and E46K) were identified in PD and DLB [7,38]. More recently, several αS pathogenic mutations, including A18T, A29S, H50Q, and G51D have been described [39]. Presumably, all of these missense mutations may be associated with protein misfolding and aggregation, resulting in increased αS evolvability during reproductive life, yet causing neurodegenerative disorders during aging. Furthermore, given the inhibitory effect of βS on aggregation of αS [40], βS might act as a buffer against αS evolvability (Figure 4a). In contrast to wild type βS, βS containing missense mutations associated with DLB, such as V70M and P123H, might promote αS aggregation [9,14], leading to increased αS evolvability (Figure 4b). Because missense mutations of βS are supposed to be beneficial in terms of αS evolvability, these mutations may survive against the pressures of natural selection. Thus, conceivably, all diseases categorized as α-synucleinopathies, such as PD, DLB, and MSA, may be regarded as antagonistic phenomena that converge on αS evolvability.

### 4.2. Sporadic DLB

Recently, several genome wide association studies (GWAS) have linked various genes to DLB. Curiously, the glucocerebrocidase (GBA) gene, which encodes a lysosomal enzyme involved in sphingolipid degradation, is deleted in Gaucher disease. It is conceivable that the linkage of Gaucher disease to sporadic PD [41] may imply that accumulated glucocerebroside due to loss of GBA function may promote αS aggregation and increase αS evolvability. Notably, it was also shown that GBA mutations are a significant risk factor for DLB, where GBA1 mutations likely play an even larger role in the genetic etiology of DLB than in PD, providing insight into the role of GBA in Lewy body disorders [42] (Figure 5). Moreover, GWAS revealed that apolipoprotein E (APOE), a major risk factor for AD, might also be a susceptibility gene linked to sporadic DLB (Figure 5) [43]. Considering that APOE ε4 binds to Aβ to stimulate fibrillization [44], it is predicted that Aβ evolvability might be enhanced by APOE ε4. Thus, although susceptibility genes in DLB are detrimental in terms of neurodegeneration in aging, they might be rather beneficial for evolvability during reproduction.

Furthermore, it is intriguing to hypothesize that alterations of wild type βS in aging, presumably through an antagonistic pleiotropy mechanism, might play a central role in DLB pathogenesis (Figure 5). As βS might create a buffering effect on AP evolvability, including both αS and Aβ, during reproduction, altering βS might enhance AP aggregation and protofibril formation, including αS and Aβ, leading to DLB in aged parents. Thus, βS might critically facilitate conversion of AP effects from evolvability to neurodegeneration through the antagonistic pleiotropy in aging.

### 4.3. Lewy Body Dementia and Amyloidogenic Evolvability

Then, by what mechanism is dementia, the major symptom in DLB, manifested? Given the association of βS with APs, including Aβ and αS, two possibilities exist. Notably, synelfin, the avian form of αS, is essential for bird song memory formation during a critical period in development [45]. If αS is also crucial for learning and memory during human neurodevelopment then, intriguingly, dementia might result from antagonistic pleiotropy phenomenon in human aging. Accordingly, it is possible that P123H βS might increase αS activity, leading to dementia. Supporting this, bigenic P123H βS and αS mice exhibited worsening neurodegeneration phenotypes (Figure 3) [14], suggesting that P123H βS may cooperate with αS to promote neurodegeneration phenotypes, including dementia, in the mouse brain.

Alternatively, cognitive deficits characterized in the P123H βS Tg mice might be derived from aggregation of endogenous AD-relevant APs, including Aβ and tau, which might be increased by expression of P123H βS. Thus, further studies are warranted to investigate such endogenous APs in P123H βS Tg mice. If cross-breeding of the P123H βS Tg mice with AD-related mice, such as either APP Tg mice or APP knockout mice, is possible, this might provide insight into such interactions. As described above, wild type βS might be altered in sporadic DLB through antagonistic pleiotropy, thus promoting AP aggregation. If correct, then familial and sporadic DLB may share essentially similar mechanisms underlying dementia. Therefore, P123H βS Tg mice might be applicable for the study of all types of DLB.

## 5. Therapy Implication

At present, no disease-modifying therapies can either arrest or slow neuronal damage in DLB. As such, current strategies focus mainly on symptomatic treatments, such as cholinesterase inhibitors, which modestly improve cognition in AD and are also approved for Lewy body dementia symptoms [46]. Furthermore, antidepressants such as selective serotonin reuptake inhibitors may be used to treat depression in DLB [47]. Moreover, clonazepam is often employed to treat REM sleep disorder [48]. Thus, in this regard, P123H βS Tg mice could be used to examine these and other symptomatic therapies to better understand their mechanisms and efficacy. In particular, non-motor symptoms in these mice, such as depression and memory dysfunction, might be ideal measures to evaluate the effects of drugs.

It is also possible that the P123H βS Tg mouse could be an ideal rodent model for examining potential disease-modifying therapies, especially more radical concepts, for DLB. As such, several potential therapy strategies are worth discussing. First, interfering with protein aggregation inhibition has been a widespread therapeutic strategy against neurodegenerative diseases, and it is noteworthy that a recent phase III Aβ immunotherapy against sporadic AD resulted in a clinically significant outcome [49]. Similarly, it is possible that reducing the burden of APs, including αS and Aβ, might be sufficient to decrease the formation of toxic amyloid fibrils and delay the progress of DLB (Figure 5: Tx1).

Second, given that αS evolvability is manifested as neurodegeneration through the antagonistic pleiotropy mechanism in aging, suppression of antagonistic pleiotropy might be therapeutically effective (Figure 5: Tx2). In this regard, transforming growth factor β/activin signaling pathways might be potentially important as therapeutic targets in terms of the antagonistic pleiotropy mechanism [36]. Additionally, provided that altering wild type βS might result in increased AP aggregation, leading to phenotypic neurodegeneration in sporadic DLB, it is possible that the modification of βS could be a therapeutic target. Future investigations are warranted to elucidate the mechanisms by which βS might be altered in aging.

Collectively, the P123H βS Tg mouse model could be useful for the development of both symptomatic and disease-modifying therapies. Given recent progress of an in vitro system using induced pluripotent stem cells (IPSCs) for the therapy of neurodegenerative diseases, including DLB [50], the effectiveness of applying IPSCs therapy to the P123H βS Tg mouse model would be of great interest. Caution must be taken, however, in recognizing the differences in aging and other effects between humans and rodents, including the antagonistic pleiotropy mechanism.

## 6. Conclusion

To summarize, the P123H βS Tg mice reproduced the many neurodegenerative features of Lewy body disease, including axonal pathology and both motor and non-motor dysfunction associated with protein aggregation. Provided with the prominent memory impairment in this mouse model, it is of great interest to determine whether the association of P123H βS with APs, including αS and Aβ, might be causally associated. According to our recent view of amyloidogenic evolvability, cross-seeding of APs, including αS and Aβ, and its negative regulation by βS, may be physiologically important for evolvability in reproduction, which could be manifested as DLB through the antagonistic pleiotropy mechanism in aging. Such concepts may be further evaluated by cross experiments of the P123H βS Tg mice with AD model mice. Furthermore, symptomatic and various disease-modifying treatments of DLB could also be assessed using these model mice, in which non-motor symptoms, such as dementia and depression, are prominent features. Thus, the P123H βS Tg mice may be a valid model for investigating the mechanism and therapy of DLB.

## Figures and Tables

**Figure 1 ijms-21-02849-f001:**
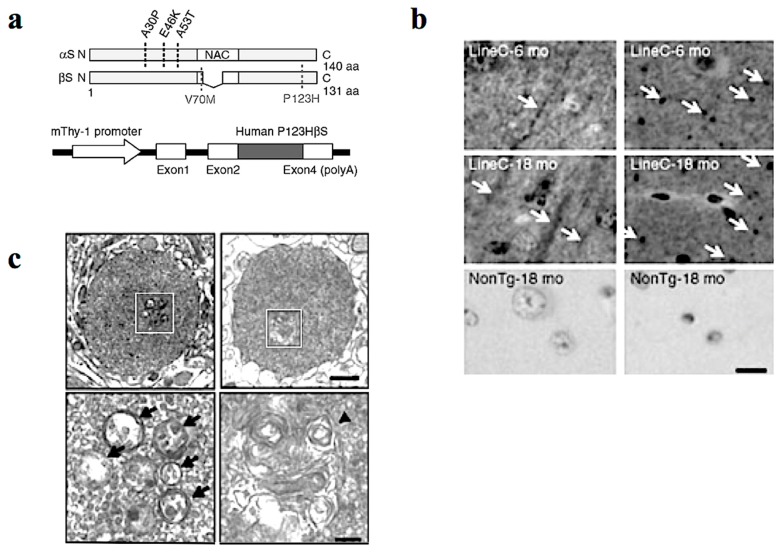
P123H βS tg mice are characterized by a neuritic pathology. (**a**) Generation of P123H βS tg mice. Schematic shows missense mutations of αS and βS identified in α-synucleinopathies. Two αS mutations, A30P and A53T, were discovered in Parkinson’s disease (PD), and another αS mutation, E46K, was later identified in dementia with Lewy bodies (DLB). As for βS, two mutations, V70M and P123H, have been reported in DLB (upper). Diagrammatic representation of the Thy-1–P123H βS construct (lower). (**b**) Immunohistochemistry of P123H βS using anti-P123H βS antibody. P123H βS was accumulated in apical dendrites in the cortex (left panels) and in axonal dots in the hippocampus (right panels) of P123H βS tg mice at 6 and 18 months (mo) (line C), but not in the same regions of non-Tg littermates (NonTg). White arrows indicate that globules were observed in the striatum and globus pallidus of 18-month-old P123H βS tg mice (lines C, arrows), but not in non-Tg littermates. Scale bar = 10 μm. (**c**) Photomicrographs of the two representative globules in P123H βS tg mice. Electron microscopy revealed that the globules were composed of small vesicles (black arrows) and frequently contained membranous elements such as multivesicular bodies and multilayered membranes (a black triangle). The specific fields surrounded by white rectangles (upper) are enlarged (lower). Scale bar = 1 μm (upper) and 200 nm (lower). Reprinted with permission from a reference [14].

**Figure 2 ijms-21-02849-f002:**
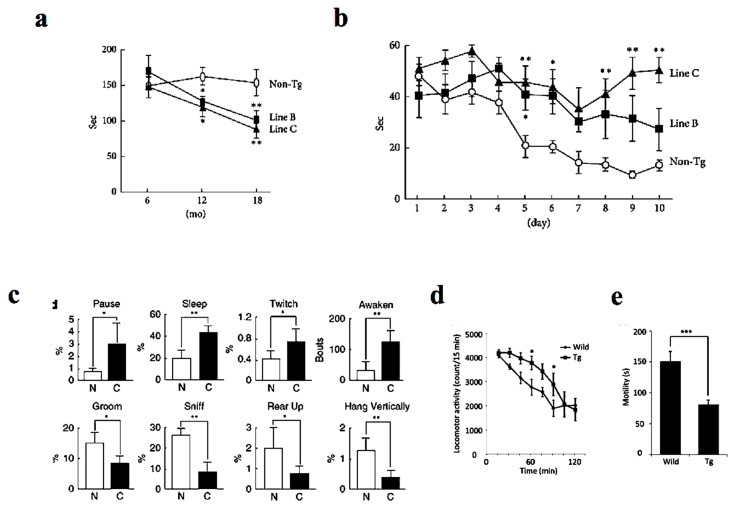
Behavioral alterations in P123H βS tg mice. (**a**) The Tg mice expressing DLB-linked P123H βS were characterized by motor dysfunction as assessed by impaired motor performance on the rota-rod treadmill test at different ages. The motor deficits become apparent after approximately 12 months. (**b**) In contrast, memory disorders were more prominent as assessed by the water maze test (apparent at approximately 6 months). (**c**) Similarly, home-cage tests showed that spontaneous activity was decreased in the P123H βS mice (at around 6 months). Furthermore, the P123H βS mice exhibited depression-like behaviors as assessed from the results of the locomotor activity (**d**) and the tai suspension test (**e**) (6–10 months). Data are shown as mean ± SEM (*n* = 8∼16). * *p* < 0.05, ** *p* < 0.01 and *** *p* < 0.001 versus non-Tg mice. Reprinted with permission from references [14,18].

**Figure 3 ijms-21-02849-f003:**
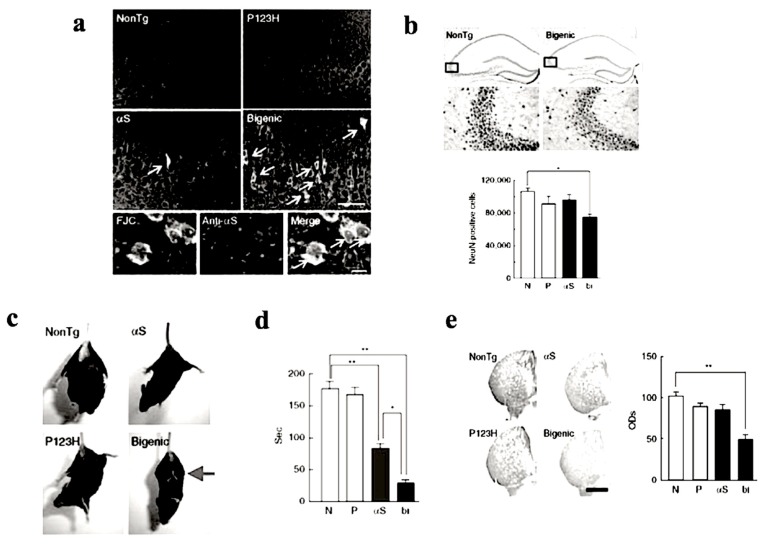
Increased nerodegeneration phenotype in bigenic (P123H βS X αS) mice. (**a**) Evaluation of neurodegeneration by Fluoro-Jade C (FJC) staining. Representative images of the hippocampus from bigenic mice and from other littermates are shown (four figures in the upper panel). FJC-positive cells were observed in bigenic mice and to a lesser extent in αS tg mice (arrows). Scale bar = 50 μm. Lower images show that FJC-stained cells were also positive for αS (arrows) in bigenic mice. Nuclei were simultaneously stained with DAPI (4,6-diamidino-2-phenylindole). Scale bar = 10 μm. (**b**) Left panels: representative images of NeuN of the hippocampus from bigenic mice and NonTg littermates are shown. Scale bar = 500 μm (upper two panels) or 100 μm (lower two panels). The figures given in the lower panels are magnifications of the figures given in the upper panel. Right panels: The graph shows neuronal density based on the NeuN-immunoreactive cell count (cells mm^−3^) in the hippocampus. Data are shown as mean ± SEM (*n* = 5). * *p* < 0.05 versus non-tg mice. (**c**) A representative photograph of the tail-suspension assay shows at 4 mo strong front and hind limb clasping in bigenic mice (arrow), but not in other littermates. (**d**) Rota-rod treadmill test shows impaired motor performance in bigenic mice and to a lesser extent in αS tg mice. Data are shown as mean ± SEM (*n* = 9–18). * *p* < 0.05, ** *p* < 0.01. (**e**) Left panels: representative images of TH immunohistochemistry at striata from 7 month bigenic mice and littermates (P123H βS tg, αS tg and non-tg) are shown. Scale bar = 500 μm. Right panels: The average optical densities (ODs) of the TH immunoreactivity was measured. Data are shown as mean ± SEM (*n* = 8). ** *p* < 0.01 versus non-tg mice. Reprinted with permission from reference [14].

**Figure 4 ijms-21-02849-f004:**
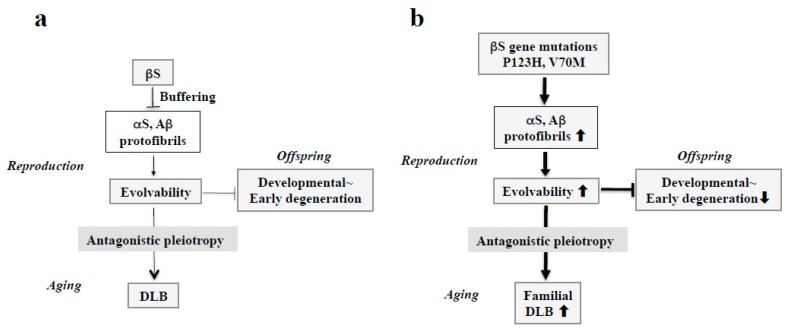
Schematics of the involvement of amyloidogenic evolvability in DLB. (**a**) Evolvability of APs, including αS and Aβ, may be an epigenetics to transmit stress information to offspring via germ cells. To regulate protein aggregation and amyloid neurotoxicity, βS may act as a buffer for this phenomenon. By virtue of this, offspring can cope with the forthcoming diverse stresses to be resistant against developmental~early degenerative diseases in offspring during the reproductive life stage. However, evolvability might become detrimental through the antagonistic pleiotropy mechanism during aging in parents, increasing the risk of DLB. (**b**) In familial DLB, βS gene mutations, including P123H and V70M, may stimulate aggregation of APs, including αS and Aβ. By virtue of the increased evolvability of APs, more information of stresses may be delivered from parents to offspring, which is beneficial for the offspring’s brain to avoid developmental~early degenerative diseases. On the other hand, increased activity of APs evolvability might be manifest as DLB through the antagonistic pleiotropy in parental aging.

**Figure 5 ijms-21-02849-f005:**
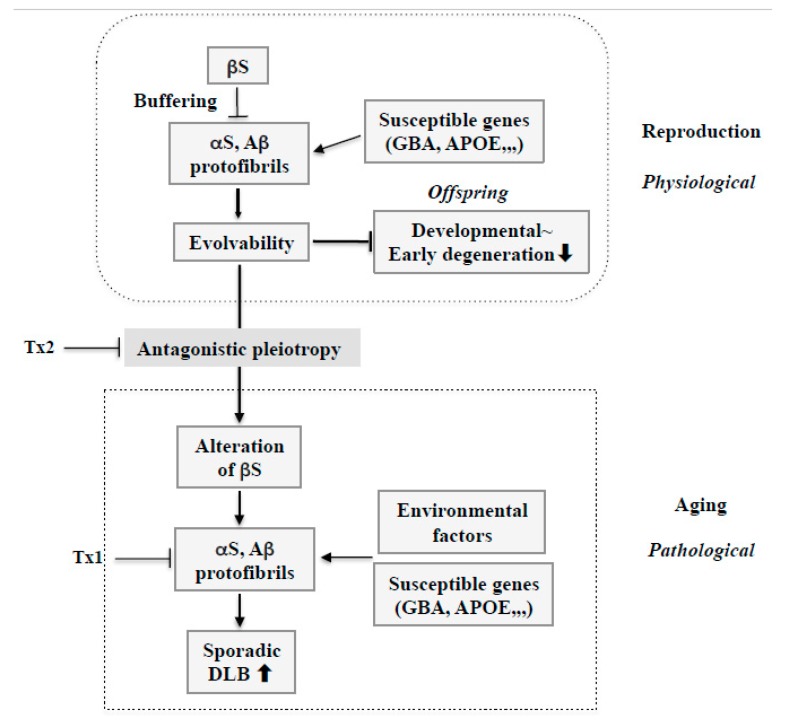
Therapy strategy against sporadic DLB based on amyloidogenic evolvability. During reproduction, stress information may be delivered by protofibrils of APs, including αS and Aβ, from parents to offspring. Therefore, increased evolvability of APs in reproduction may be beneficial for the offspring’s brain to avoid developmentally early degenerative diseases. Evolvability of APs, however, becomes detrimental through the antagonistic pleiotropy mechanism in aging. βS might play an important role as a buffer for evolvability of APs. However, βS might be altered through the antagonistic pleiotropy mechanism in aging and stimulate formation of protofibrils of APs. In this context, susceptible genes, such as GBA and APOE, might be beneficial for evolvability in reproduction, but may stimulate the neurotoxic protofibrils formation of APs combined with environmental factors, leading to neurodegeneration in aging. Based on such a view, at least two therapy strategies against sporadic DLB could be considered. First, given the importance of APs protofibrils in neurodegeneration in aging, dose-reduction therapy should be considered for APs, including αS and Aβ (Tx1). Second, neurodegeneration by APs protofibrils in aging might be promoted by the antagonistic pleiotropy mechanism. Therefore, this process could be a therapeutic target (Tx2).

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
