# Peer review of "Possible Role of Amyloidogenic Evolvability in Dementia with Lewy Bodies: Insights from Transgenic Mice Expressing P123H β-Synuclein"

_ijms, 2020, doi:10.3390/ijms21082849_

Round 1

Reviewer 1 Report

Topic of this review trying to cover one of challenging area in the neurodegenerative diseases topic. Because until today no treatment option are available for Alzheimer or other type dementia including Lewy Bodies.

Unfortunately this review lack of scientific novelty and below some of key missing point that will be able to provide for authors' guideline to improve this draft.

I am confident that authors’ very familiar failure of amyloid based theory of Alzheimer as well as any other dementia. All of drugs that developed based on the amyloid theory already failed and I am confident that this is also appeared to be one of blocking factor for the development of new strategies  regarding diseases pathobiology and drug development. Authors’ should consider recent strong evidence all of new and promising theory including oxidative stress based theory of dementia related neurodegeneration. Even data presented in this review also indicates that neuronal pathology become first and amyloid accumulation appeared to be as a consequence but not reason for neuronal pathology.

Therefore at the present form this manuscript cannot be considered as a completed study.

Reviewer 2 Report

Dear authors, 

This is an interesting paper, however I don't know if the denomination of "review" is the correct one. The paper develops a hypothesis based on a compilation of your previous studies. A review will be a compilation of several works of a topic without formulating any hypothesis, simply explaining the current state-of-the-art of the topic. 

I will suggest you define well your paper. I assume figures without this sentence "Reprinted with permission from references" came from new research. Thus, if you have a new data, you should submit as original article. I also suggest removing some figures because you can refer to the article. 

Thank you very much. 

Round 2

Reviewer 1 Report

Unfortunately revised manuscript does not contain any scientific novelty.

Adding just one paragraph not going to make any differences regarding the scientific novelty of this manuscript  that can be called self review.